# COVID-19 and non-Hodgkin's lymphoma: A common susceptibility pattern?

**De Matteis Sara[1,2], Cosetta Minelli[2], Giorgio Broccia[3], Paolo Vineis[4], Pierluigi Cocco [5]***

**1** Department of Medical Sciences and Public Health, University of Cagliari, Cagliari, Italy, **2** National Heart and Lung Institute, Imperial College London, London, United Kingdom, **3** Private Consultant, Former director of the Department of Haematology and Bone Marrow Transplants, Hospital A. Businco, Cagliari, Italy, **4** Faculty of Medicine, School of Public Health, Imperial College, London, United Kingdom, **5** Centre for Occupational and Environmental Health, Division of Population Health, University of Manchester, Manchester, United Kingdom

\* pierluigi.cocco@manchester.ac.uk

**Data Availability Statement:** Data are stored on the figshare repository and are publicly available (DOI: 10.6084/m9.figshare.22210675

**Funding:** The authors received no specific funding for this work.

## Abstract

### Objective

To explore the link between COVID-19 incidence, socio-economic covariates, and NHL incidence.

### Design

Ecological study design.

### Setting

Sardinia, Italy.

### Participants

We used official reports on the total cases of COVID-19 in 2020, published data on NHL incidence, and socio-economic indicators by administrative unit, covering the whole regional population.

### Main outcomes and measures

We used multivariable regression analysis to explore the association between the natural logarithm (ln) of the 2020 cumulative incidence of COVID-19 and the ln-transformed NHL incidence in 1974–2003, weighing by population size and adjusting by socioeconomic deprivation and other covariates.

### Results

The cumulative incidence of COVID-19 increased in relation to past incidence of NHL ($p < 0.001$), socioeconomic deprivation ($p = 0.006$), and proportion of elderly residents ($p < 0.001$) and decreased with urban residency ($p = 0.001$). Several sensitivity analyses confirmed the finding of an association between COVID-19 and NHL.

**Competing interests:** The authors have declared that no competing interests exist.

## Conclusion

This ecological study found an ecological association between NHL and COVID-19. If further investigation would confirm our findings, shared susceptibility factors should be investigated among the plausible underlying mechanisms.

## Introduction

The worldwide geographical variation in the incidence of and mortality from the 2019 coronavirus disease (COVID-19) and the inter-individual susceptibility to its more severe forms suggests a potential interplay of genetic susceptibility and environmental factors [1]. Ecological research has suggested possible contributions from population density and vehicular traffic [2], temperature and humidity [3], and air pollution [1, 4], and an inverse association with the vaccination against seasonal influenza during the first epidemic wave but not afterward [2, 5]. Among the genetic variants conferring increased susceptibility to SARS-CoV-2 infection, those expressing the Angiotensin-Converting Enzyme (ACE) receptors [6] are key-factors in the SARS-CoV-2 cell membranes' crossing, and HLA-DRB1 alleles were more frequently observed in symptomatic COVID-19 patients [7]. Also, the cytokine storm, a feature of the more severe forms of COVID-19, is reflected in a sharp increase in interleukin 10 (IL-10) [8] and TNF-$\alpha$ [9] serum levels, which would depend on the expression of the respective genes. Furthermore, specific TNF-$\alpha$ and TNF-$\beta$ gene polymorphisms increase the risk of COVID-19 occurrence and severity [10]. A few IL-10, IL17A, and IL17F single gene polymorphisms (SNP), but not the other IL-10 SNPs, have also been shown to affect the prevalence and clinical outcome of the disease [11, 12]. Besides, the Janus kinase (JAK/STAT) gene expression would modulate the SARS-CoV-2 induction of the complement activation [13], and. the decrease in the 2'-5'-oligoadenylate synthase (OAS1) gene expression would contribute to COVID-19 severity [14].

Interestingly, the genetic factors associated with the appearance of symptoms and the unfavourable evolution of COVID-19 are also implicated in the aetiopathogenesis of non-Hodgkin's lymphoma (NHL). For instance, the risk of various lymphoma subtypes increases in association with HLA class I and class II variants [15–17] and polymorphisms in genes expressing IL-10 and TNF [18] and others related to the JAK-STAT signalling pathway [19]. The apparent resemblance between the genetic features characteristics of the more severe forms of COVID-19 and those implicated in the NHL pathogenesis matches the link between previous viral and microbial infections, including HIV infection, and NHL risk, whether through immunosuppression or other mechanisms [20, 21]. In agreement with the hypothesis of shared pathways, a few reports have described an increase in COVID-19 incidence and severity among patients with a diagnosis of primary cutaneous lymphoma (PCL) [22, 23] and haematological disease in general [24, 25] but not among cancer patients, including lymphoma patients [26]. However, patients with specific NHL subtypes may differ in terms of vulnerability to SARS-CoV-2 infection; it is unclear whether treatment-induced immunosuppression [27, 28] and the generic low-rate seroconversion in onco-haematological patients [29] would be exhaustive explanations for such differences.

A recently published Bayesian analysis showed that, over the three decades between 1974 and 2003, the NHL incidence tended to cluster in the north-eastern part of the Italian region of Sardinia and its major urban centre, with the low incidence areas located in the south [30]. The ecological analysis did not identify plausible explanatory factors among the conditions

known to increase NHL risk. If common susceptibility factors contributed to both COVID-19 and NHL, one would expect the two diseases to show consistent geographic patterns. We used the 2020 cumulative incidence data for COVID-19 by administrative units to explore the correlation between the past incidence of NHL and the incidence of COVID-19 within Sardinia.

## Methods

The age- and gender-standardized incidence rates of NHL between 1974–2003 were available for all the 356 autonomous administrative units (communes) in Sardinia as of 1974 [30]. Briefly, due to the lack of a regional Cancer Registry, the chief haematologist of the Cagliari Oncology Hospital, with the support of the health authorities and the clinical departments of the whole region, created and regularly updated a database by prospectively registering all incident haematological cancers diagnosed among the Sardinian population of both genders and any age from 1974–2003 [31]. We validated the completeness of the database records by comparing the resulting incidence rates with region-wide mortality and hospitalization data [32]. Cancer Registry data became available limited to the last decade and the northern part of the region; the comparison with these data also confirmed the completeness of the database we used [33]. For each commune, we abstracted data on the cumulative incidence of COVID-19 from the first diagnosis to 31 December 2020, before the start of the vaccination campaign, from a publicly available report [34]. Only the crude cumulative incidence of COVID-19 was locally available. Also, we could not analyse COVID-19 mortality nor the rate of positives over the total tests performed or the disaggregated figures by symptomatic or asymptomatic positive cases, as such information was unavailable at the commune level. The communes' population varied from 77 to 195,500, with 196 (55%) inhabited by less than 2,000 people and 26 (7%) comprising 56% of the resident population. Despite being the most prevalent group of onco-haematological neoplasms, NHL is relatively rare, with a global incidence of 7.8 per 100,000 per year (all ages) [35]. Therefore, the precision of the estimate of the local NHL incidence rate was very low for many little communes. For instance, 25 communes (all with less than 2,000 inhabitants) had no incident NHL cases in 1974–2003, while the incidence varied from 1.2 to 17.4 per 100,000 in the 74 communes where the cases were only 1 or 2. The 2020 incidence of COVID-19 manifested the same zero-case occurrence in four communes with less than 500 inhabitants. We graphically addressed this problem using bubble plots, with the bubble radius proportional to the commune's population size, and analytically by conducting a weighted regression using the ratio between the local and the regional population as the weight.

We repeated the analysis with natural log-transformed (ln) age- and sex-standardised NHL incidence rates (independent variable) and the COVID-19 incidence rates (outcome), as the residuals of the regression model using the raw data violated normality at the lower and upper tails. In this analysis, we added a small constant to each value of both variables to allow the ln transformation even for zero values. In a further analysis, we used conventional (instead of a weighted) regression analysis after grouping the communes with less than 10,000 residents into 36 larger units corresponding to historical sub-regional areas and the NHL incidence in these and the remaining 26 larger communes as the independent variable. This analysis comprised 61 units instead of 62 because one of the 26 larger communes was part of an area consisting solely of two communes. We also set multiple regression models to explore socio-economic covariates possibly confounding the association, including the proportion of inhabitants aged ≥75 years, the male/female ratio among the residents, the Italian Institute of Statistics (ISTAT) deprivation index (http://istat.it), the distance from the nearest hospital, and the urban/rural type of commune. The urban or rural type covariate for each commune was defined based on the presence/absence of five community services (administrative,

educational, health, judicial, and religious) that would attract daily commuters from the surrounding area. For each aggregated area, the deprivation index, the proportion of elderly, and the male/female ratio were calculated as weighted averages. Further analyses included 1) considering only the communes with more than 2,000 residents and 2) conducting the regression analysis within each approximate quartile of the distribution of the communes' resident population.

The best-fitting model was selected using a stepwise backward approach, with the goodness-of-fit of the models assessed using the $R^2$ value, which indicates the proportion of the variability in the COVID-19 incidence rate explained by the covariates in the model. The analysis was conducted using SPSS® version 20.0.

A STROBE statement on ecological studies, such as the present study, is still missing. However, we acknowledged the limitations of such a study design and complied with the requirements for ecological studies. In our analysis, we only accessed publicly available data and aggregated data, which use for scientific publications was approved by the Ethics Committee of the University Hospital of Cagliari (protocol N. PG 2019/18070, 18 December 2019).

## Results

The 1974–2003 NHL age- and gender-adjusted incidence rate among the Sardinian adult population (aged 25 years or more) was 13.4 per 100,000 (95% CI 13.0–13.8) [30], ranging from 0 to 44.4 across the 356 communes. The 2020 cumulative incidence of COVID-19 amongst the regional population was 2,117 per 100,000 (95% confidence interval [CI] 2,095–2.137), ranging from 0 in four communes with 77, 148, 224, and 506 residents, respectively, to 11,935 in a commune with 1,240 residents, based on 148 incident cases (12% of the total residents). Fig 1 shows the bubble plots of the COVID-19 incidence against NHL incidence. The regression line in Fig 1A describes the weighted regression equation in all the communes, and that in Fig 1B describes the unweighted regression equation in the 61 units (25 large communes and 36 aggregated areas); both show an upward trend of increasing 2020 COVID-19 incidence with increasing past NHL incidence. The analysis of ln-transformed data yielded similar results (data not shown). Using the incidence rate among subjects aged 45 or more, an age range closer to that of the vast majority of COVID-19 cases during the 2020 epidemic, did not substantially change the correlation coefficient (NHL incidence rate among 25+ years old: $r = 0.642$, $p < 0.001$; among 45+ years old: $r = 0.647$, $p < 0.001$).

The best-fitted multiple regression model (adjusted $R^2 = 0.31$) included the ln-transformed NHL incidence rate ($\beta = 1.021$, $p < 0.001$), the proportion of elderly among the resident population ($\beta = 0.427$, $p < 0.001$), the type of residence ($\beta = -0.331$ for urban *vs* rural residence, $p = 0.008$), and the deprivation index ($\beta = 0.134$, $p = 0.035$) as the covariates (Table 1, Model 1). The distance to the nearest hospital and the male/female ratio were not associated with COVID-19 incidence, nor did their inclusion in the model result in a better fit. The analyses of the 61 administrative units (Table 1, Model 2) and by gender (data not shown) confirmed the results.

Further sensitivity analyses, by dropping the small size communes (<2,000) and by quartile of population size, consistently showed an increase in COVID-19 cumulative incidence in relation to past NHL incidence (data not shown).

## Discussion

Our results suggest a significant geographic correlation between the 2020 cumulative incidence of COVID-19 and the 1974–2003 NHL incidence within the region of Sardinia. We limited the analysis to the first year of the COVID-19 pandemic, before vaccines became available, to

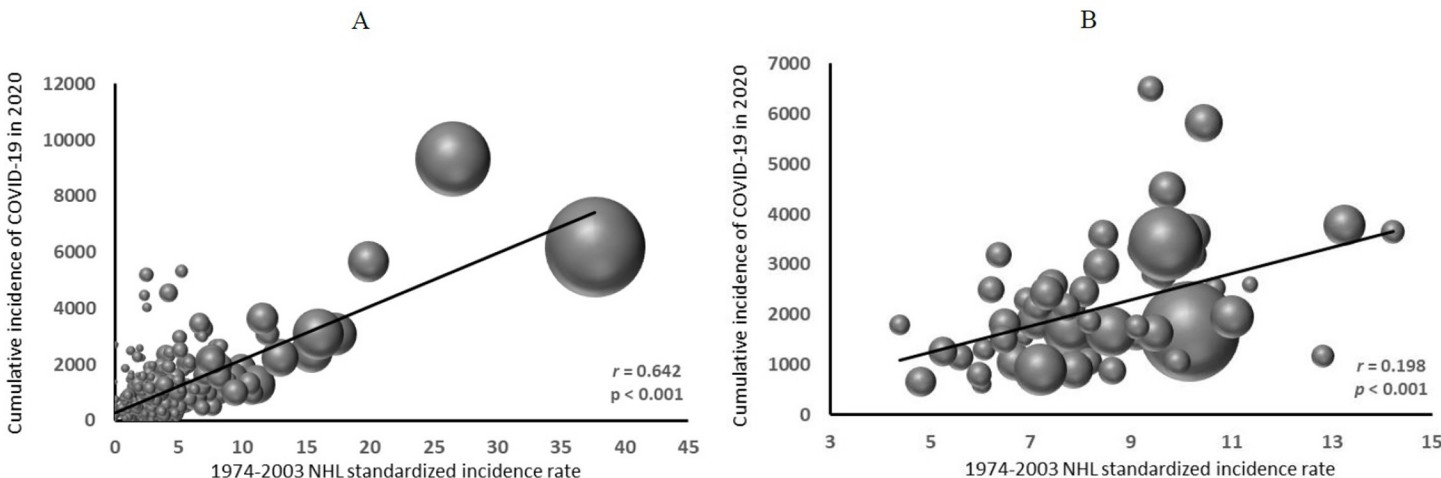

**Fig 1. Bubble graph of 2020 cumulative incidence of COVID-19 in relation to 1974–2003 incidence of non-Hodgkin's lymphoma in Sardinia, Italy.** A. all communes (N = 356), weighed (weigh = resident population); B. 61 residential units, including 25 communes with 10000 residents or more and 36 sub-regional areas incorporating the remaining 331 communes. Each commune is represented by a bubble of diameter proportional to its population size relative to the total regional population.

avoid the effect of the geographical and age-related variation in compliance with the vaccination. The analysis restricted to the NHL incidence rate in subjects aged 45 years or older confirmed the result. To overcome the analytical problem created by the sparse data in the numerous small-size communes, we conducted additional analyses by aggregating the smaller communes, by stratifying the analysis by commune size, and on ln transformed data: all the regression analyses confirmed the strong correlation between the past incidence of NHL and the 2020 cumulative incidence of COVID-19. Based on the weighted regression analysis, a 1% increase in the past NHL incidence would have yielded a 1.02% increase in the incident cases of COVID-19 per 100,000 residents in 2020, corresponding to 22 additional cases. Amongst the other variables included in our model, consistently with previous reports, the proportion of elderly residents and socio-economic deprivation also showed an association with an increasing cumulative incidence of COVID-19 [1, 36]. On the other hand, sex and distance from the nearest hospital did not show an association, and urban residency was inversely related to COVID-19 incidence.

Only the total number of incident COVID-19 cases was publicly accessible at the local level, which did not allow a proper age and gender standardization of the rates. This drawback limits the interpretation of our results; to mitigate the resulting bias, we tested the correlation with the NHL incidence at age 45 years or older, an age range closer to that most frequently

**Table 1. Parameters of the multiple regression analysis with cumulative incidence of COVID-19 as the outcome: Model 1: Weighted regression, all communes; outcome: ln-transformed COVID-19 incidence; covariates: ln-transformed standardized NHL incidence, deprivation index, proportion of elderly residents, and urban vs rural residence.** The last three are not ln-transformed; model 2: conventional regression, 61 administrative units: outcome: cumulative incidence of COVID-19; covariates: standardized NHL incidence, deprivation index, proportion of elderly residents.

| Independent covariates | Model 1 | | | Model 2 | | |
|---|---|---|---|---|---|---|
| | β | se | p | β | se | p |
| Ln-transformed NHL incidence | 1.021 | 0.196 | < 0.001 | 244.6 | 69.2 | < 0.001 |
| Deprivation index | 0.134 | 0.063 | 0.035 | 93.8 | 116.4 | 0.424 |
| Proportion of elderly residents | 0.427 | 0.101 | < 0.001 | 288.0 | 120.5 | 0.020 |
| Urban vs rural residence | -0.331 | 0.124 | 0.008 | - | | |
| Adjusted R² | 0.30 | | | 0.25 | | |

represented among the COVID-19 cases: the correlation coefficient was similar. In the multiple regression models, we used the proportion of elderly and the male/female ratio among the residents as adjusting covariates. While the male/female ratio did not show an association with COVID-19 incidence nor contributed to reducing the model variance, the proportion of elderly did. As incidence increased nationwide among the elderly but varied little by gender (4.7% of the male population vs 4.6% of the female population) [37], we presumed to have at least partially rectified the drawback. A further limitation results from the unavailability of the number of swab tests at the regional and commune level, which would have allowed using a perhaps more reliable measure of incidence. Nationwide, molecular testing was widely accessible to the general public as soon as it became available. It was initially used as a diagnostic tool among symptomatic subjects and in the health surveillance of the healthcare staff and was lately required for traveling and accessing public offices. In Sardinia, the Regional Administration also funded a COVID-19 testing program open to the general public on a voluntary basis. At the national level, the nasopharyngeal swab rate increased steadily during the first epidemic wave from 1 March to 13 May (conventionally set as the end of the first epidemic wave based on the nadir of the epidemic curve), and, during the second epidemic wave (conventionally set between 13 September and 28 December) remained stable up to the middle of December when it started to decline [5]. Mortality data and disaggregated figures of COVID-19 incident cases by the presence or absence of symptoms were also unavailable; therefore, we could not investigate the most severe forms of COVID-19.

The suspicion of bias due to the "ecological fallacy" has repeatedly been raised for findings in ecological studies, as associations at the population level might not translate into associations at the individual level [38]. However, ecological studies are helpful in the preliminary exploration of the link between new health emergencies, such as the global COVID-19 pandemic, and possible pre-existing conditions and risk factors in a population. Such studies can provide aetiological clues to direct further research aimed at informing public health preventive strategies [39]. Therefore, while recommending further epidemiological investigations into the association between NHL and COVID-19, we caution against over-interpreting our findings.

If further research confirmed our findings, a shared susceptibility between NHL and COVID-19 might be postulated, including genetic susceptibility. If so, polymorphisms in genes implicated in the synthesis of cytokines, HLA cell surface antigens, and others facilitating the virus entry into human cells or impairing the immune response might play a role in the pathogenesis of both diseases. Consequently, the greater susceptibility of lymphoma patients to SARS-CoV-2 infection and the most severe forms of COVID-19 might be due not only to the immunosuppressive effects of the therapy protocols but also to a common susceptible background for NHL and COVID-19.

## Acknowledgments

The authors are grateful to Mr Maurizio Addis and Mr Franco Porcu for local advice and support in accessing the official COVID-19 incidence data.

## Author Contributions

**Conceptualization:** De Matteis Sara, Pierluigi Cocco.

**Data curation:** De Matteis Sara, Giorgio Broccia.

**Formal analysis:** De Matteis Sara, Cosetta Minelli, Pierluigi Cocco.

**Investigation:** De Matteis Sara, Giorgio Broccia.

**Methodology:** Cosetta Minelli, Paolo Vineis.

**Supervision:** De Matteis Sara, Paolo Vineis.

**Validation:** Paolo Vineis, Pierluigi Cocco.

**Visualization:** Pierluigi Cocco.

**Writing – original draft:** De Matteis Sara, Pierluigi Cocco.

**Writing – review & editing:** De Matteis Sara, Cosetta Minelli, Giorgio Broccia, Paolo Vineis, Pierluigi Cocco.

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
