## [Decision Letter · Decision Letter 0]

18 Jan 2023

PONE-D-22-29855COVID-19 and non-Hodgkin’s lymphoma: a common susceptibility pattern?PLOS ONE

Dear Dr. Pierluigi Cocco,

Thank you for submitting your manuscript to PLOS ONE. After careful consideration, we feel that it has merit but does not fully meet PLOS ONE’s publication criteria as it currently stands. Therefore, we invite you to submit a revised version of the manuscript that addresses the points raised during the review process.

ACADEMIC EDITOR: Upon reviewing this manuscript, I like the justification provided by the authors for correlating COVID-19 disease with NHL. The manuscript is well-written. However, I concur with the concerns raised by the reviewer and urge the authors to address each of these before resubmission.

We look forward to receiving your revised manuscript.

Kind regards,

M. Kariuki Njenga

Academic Editor

PLOS ONE

Journal Requirements:

2.We note that you have included the phrase “data not shown” in your manuscript. Unfortunately, this does not meet our data sharing requirements. PLOS does not permit references to inaccessible data. We require that authors provide all relevant data within the paper, Supporting Information files, or in an acceptable, public repository. Please add a citation to support this phrase or upload the data that corresponds with these findings to a stable repository (such as Figshare or Dryad) and provide and URLs, DOIs, or accession numbers that may be used to access these data. Or, if the data are not a core part of the research being presented in your study, we ask that you remove the phrase that refers to these data.

6. Please note that in order to use the direct billing option the corresponding author must be affiliated with the chosen institute. Please either amend your manuscript to change the affiliation or corresponding author, or email us at plosone@plos.org with a request to remove this option. 

Reviewers' comments:

Reviewer's Responses to Questions

**Comments to the Author**

1. Is the manuscript technically sound, and do the data support the conclusions?

Reviewer #1: Partly

2. Has the statistical analysis been performed appropriately and rigorously? 

Reviewer #1: No

3. Have the authors made all data underlying the findings in their manuscript fully available?

Reviewer #1: No

4. Is the manuscript presented in an intelligible fashion and written in standard English?

Reviewer #1: Yes

5. Review Comments to the Author

Reviewer #1: General comments

The authors do not give enough background information on COVID-19 testing for the reader to understand what the limitations of the COVID-19 data are. Was testing widely available? Was it limited to those who had symptoms only? Was there massive uptake of testing by the general public?

The limitations of ecological studies have been noted by the authors which I agree with. My main concern is that the rates of NHL and COVID-19 incidence refer to two different age groups - the details of which I have provided below.

Introduction

“Also, during the first epidemic wave but not afterwards, the vaccination against seasonal influenza provided protection – the authors appear to state as a fact that influenza vaccination protects against SARS-CoV-2 infection. This is not true. The reference stated is an ecological study which is weak evidence at best. The sentence should be reframed to make clear what the evidence is regarding this statement.

The message is the first paragraph appears not to be fully formulated. It is not clear to me why the authors mention findings from ecological studies in passing then immediately pivot to genetic variants that influence COVID-19 infection and severity. What is the relevance of mentioning the ecological studies? Did the authors mean to link the ecological studies to the genetic variants? Are they two distinct messages? If yes, the message on ecological studies is incomplete – what is the relevance of these ecological studies to this study? The issues on genetic variations move from susceptibility to infection to severe disease then back to infection. It would be easier to read if the authors were systematic in their presentation - focusing on susceptibility to infection then susceptibility to symptomatic or severe disease. However, the authors do not provide the relevance of mentioning severe disease and how it relates to their own paper. Is it that more severe disease is more likely to be detected and that’s why they mention it. If there is no link between more severe COVID-19 disease and the objectives of their study then the information is distracting and not essential to the manuscript.

“However, NHL subtypes may differ in terms of vulnerability to SARS-CoV-2 infection. Treatment-induced immunosuppression [26,27] and a low-rate seroconversion [28] in onco-haematological patients would only partly explain such differences.” – seroconversion to what? Do the authors mean seroconversion following SARS-CoV-2 infection. If that is the case are they referring to vulnerability to SARS-CoV-2 re-infection? The message the authors are relaying is not clear.

“Taking profit” – this is not a commonly used term. Consider using a different term.

“COVID-19 has shown a direct link with TNF-� and TNF-� gene polymorphisms” – what type of link is the author referring to? The reader is forced to open up the references to understand the nature of this link.

Methods

“Briefly, due to the lack of a regional Cancer Registry, the chief haematologist of the Cagliari Oncology Hospital, with the support of the health authorities and the clinical departments of the whole region, created a database of all incident haematological cancers, diagnosed among the Sardinian population of both genders and any age from 1974-2003 [30].” – when was this done? Has the data been collected prospectively from 1974 or was this done retrospectively.

“Its completeness was validated through the comparison with mortality and hospitalization data [31] and, limited to the last decade and the northern part of the region, with Cancer Registry records [32].” – this sentence is not easily understood.

Analysis

It is not clear to me what a sensitivity analysis by quartile of population size is.

The incidence of NHL is presented among individuals 25 years of age and older for the period 1974-2003 but COVID-19 incidence is presented across all ages. Shouldn’t the analysis have been done using the same age groups if the authors intention is to imply that those who have NHL are at higher risk of COVID-19 infection i.e. the COVID-19 incidence should have been limited to those 42 years of age and older.

Results

The author implies in the first paragraph that the incidence of COVID-19 was 11,935 per 100,000 population in a commune with 1,240 which seems impossible. Is this accurate?

Why are there no confidence intervals round the COVID-19 incidence rates?

6. PLOS authors have the option to publish the peer review history of their article (what does this mean?). If published, this will include your full peer review and any attached files.

Reviewer #1: No

---

## [Author Response · Author response to Decision Letter 0]

7 Feb 2023

Reviewer #1: General comments

1. “The authors do not give enough background information on COVID-19 testing for the reader to understand what the limitations of the COVID-19 data are. Was testing widely available? Was it limited to those who had symptoms only? Was there massive uptake of testing by the general public?”

In the previous version, we mentioned this point as a limitation. The revised version (unmarked version of the revised version, discussion section, second paragraph, last four lines on page 10 -through line 6 on page 11) includes a new paragraph explaining that molecular testing was widely available to the general public nationwide and it was mandatory for air and train traveling and to work on-site and access hospitals and public offices. It was also performed on a voluntary basis by anyone who had symptoms and anyone willing to take it by effect of a specific COVID-19 testing program funded by the Regional Administration. Data on nasopharyngeal swabs were unavailable at the regional level. At the national level, the swab rate increased steadily during the first epidemic wave from 1 March to 13 May (conventionally set as the end of the first epidemic wave based on the nadir of the epidemic curve). During the second epidemic wave, conventionally set between 13 September and 28 December, the swab rate remained stable through the middle of December and started declining afterwards. Please refer to Cocco P, De Matteis S. Epidemiol Infect 2022:1-26. https://doi: 10.1017/S095026882200084X) for details (reference # 5 in the list of references of the revised version). 

2. “The limitations of ecological studies have been noted by the authors which I agree with. My main concern is that the rates of NHL and COVID-19 incidence refer to two different age groups.” “The incidence of NHL is presented among individuals 25 years of age and older for the period 1974-2003 but COVID-19 incidence is presented across all ages. Shouldn’t the analysis have been done using the same age groups if the authors intention is to imply that those who have NHL are at higher risk of COVID-19 infection i.e. the COVID-19 incidence should have been limited to those 42 years of age and older.”

We thank the reviewer for this insightful comment. In the previous version, we acknowledged the difference between the NHL incidence rates (ages 25+, age- and gender-adjusted) and the COVID-19 cumulative incidence (crude) as a limitation due to the unavailability of the incident COVID-19 cases by age and gender. In the revised version, we followed the reviewer’s suggestion and calculated the NHL incident rate among individuals aged 45 years or older to make the two rates more comparable. Figure 1 shows the result of the regression analysis.

The correlation between the past incidence of NHL and the 2020 cumulative incidence of COVID-19 did not substantially change when using the NHL incidence at age 45 or older. In the revised version, we mentioned this new analysis in the results section and the discussion.

Figure 1. Graph plotting the cumulative incidence of COVID-19 against the 1974-2003 NHL incidence rate at age 25+ (A) and the same incidence rate at age 45+ (B). Each commune is represented by a bubble of diameter proportional to its population size relative to the total regional population. 

A

B

3. Introduction. “...the authors appear to state as a fact that influenza vaccination protects against SARS-CoV-2 infection. This is not true. The reference stated is an ecological study which is weak evidence at best. The sentence should be reframed to make clear what the evidence is regarding this statement.”

In the revised version, we rephrased the sentence mentioned by the reviewer and clarified that the evidence came from ecological studies.

4. “The message is the first paragraph appears not to be fully formulated. It is not clear to me why the authors mention findings from ecological studies in passing then immediately pivot to genetic variants that influence COVID-19 infection and severity. What is the relevance of mentioning the ecological studies? Did the authors mean to link the ecological studies to the genetic variants? Are they two distinct messages? If yes, the message on ecological studies is incomplete – what is the relevance of these ecological studies to this study?” 

The first sentence of our paper cites ref. # 1 suggesting an interplay between genetic susceptibility and environmental factors might explain the geographic variability of COVID-19 incidence. Many scholars adopted the ecological approach as soon as official data were made available to investigate the possible contribution of environmental variables to the occurrence of the infection. These are still the only studies addressing the hypothesis. The paragraph that follows describes the genetic factors. In the revised version (unmarked version, introduction, lines 1-6), we modified the sentence to better describe the current knowledge on the factors that contribute to the geographic variation in COVID-19 incidence.

5. “The issues on genetic variations move from susceptibility to infection to severe disease then back to infection. It would be easier to read if the authors were systematic in their presentation - focusing on susceptibility to infection then susceptibility to symptomatic or severe disease. “

In the revised version, we reorganized the paragraphs presenting the current state of knowledge on the genetic factors associated with the susceptibility to the infection, the appearance of symptoms , and the more severe forms of the disease.

6. “...the authors do not provide the relevance of mentioning severe disease and how it relates to their own paper. Is it that more severe disease is more likely to be detected and that’s why they mention it. If there is no link between more severe COVID-19 disease and the objectives of their study then the information is distracting and not essential to the manuscript.”

In our view, the appearance of symptoms and the unfavourable evolution of COVID-19 are major drivers for testing and diagnosing the disease. In the revised version (unmarked version, introduction, page 4, lines 19-23), we now explain that the genetic factors associated with the more severe forms have also been implicated in the pathogenesis of NHL.

7. “However, NHL subtypes may differ in terms of vulnerability to SARS-CoV-2 infection. Treatment-induced immunosuppression [26,27] and a low-rate seroconversion [28] in onco-haematological patients would only partly explain such differences.” – seroconversion to what? Do the authors mean seroconversion following SARS-CoV-2 infection. If that is the case are they referring to vulnerability to SARS-CoV-2 re-infection? The message the authors are relaying is not clear.”

In the revised version (unmarked copy, introduction, page 5, lines 6-9), we modified this sentence and the preceding paragraph to clarify the hypothesis of shared pathways between NHL and infections in general and COVID-19 in particular.

8. “Taking profit” – this is not a commonly used term. Consider using a different term.”

We rephrased the sentence according to the reviewer’s suggestion.

9. “COVID-19 has shown a direct link with TNF-� and TNF-� gene polymorphisms” – what type of link is the author referring to? The reader is forced to open up the references to understand the nature of this link.”

In the revised version (unmarked version, introduction, page 4, lines 21-23), we clarified the effect of TNF-� and TNF-� genetic variants on the prevalence and severity of COVID-19. This point is incorporated in the revised version of the whole paragraph described in point # 6.

10. “Methods. Briefly, due to the lack of a regional Cancer Registry, the chief haematologist of the Cagliari Oncology Hospital, with the support of the health authorities and the clinical departments of the whole region, created a database of all incident haematological cancers, diagnosed among the Sardinian population of both genders and any age from 1974-2003 [30].” – when was this done? Has the data been collected prospectively from 1974 or was this done retrospectively.”

One of the co-authors (GB) started creating the register of all incident diagnoses of haematological malignancies since he became the director of the Oncohaematology department of the Oncology Hospital that became operative in 1974. He kept updating regularly the database along the years up to retirement in 2003. In the revised version, we clarified this point. 

11. “Its completeness was validated through the comparison with mortality and hospitalization data [31] and, limited to the last decade and the northern part of the region, with Cancer Registry records [32].” – this sentence is not easily understood.”

we modified the sentence as the reviewer requested (unmarked copy of the revised version, methods section, page 6, lines 1-4). We hope that now it more clearly explains the validation procedure.

12. “Analysis. It is not clear to me what a sensitivity analysis by quartile of population size is.”

We used the term “sensitivity analysis” to indicate that we repeated the correlation analysis within strata of the communes based on the quartile distribution of the resident population. In the revised version, we rephrased this sentence to make it clearer.

13. “Results. The author implies in the first paragraph that the incidence of COVID-19 was 11,935 per 100,000 population in a commune with 1,240 which seems impossible. Is this accurate?”

In that commune, in 2020, 148 COVID-19 cases were diagnosed among 1,240 residents (11.9%). It was the highest cumulative incidence among the Sardinian administrative units. Because of the wide range of the population size across the communes, and to use the same unit for COVID-19 cumulative incidence and incidence rate of NHL, we set to 100,000 the denominator for both rates, which is why the cumulative incidence was so high.

14. “Why are there no confidence intervals round the COVID-19 incidence rates?”

As the number of COVID-19 cases referred to the whole regional population and not to a sample we thought the 95% confidence interval was unnecessary. However, as the reviewer thinks differently and having reported it for the NHL incidence rate, in the revised version, we now report the 95% confidence interval of COVID-19 cumulative incidence.

---

## [Editor Report · Decision Letter 1]

27 Feb 2023

COVID-19 and non-Hodgkin’s lymphoma: a common susceptibility pattern?

PONE-D-22-29855R1

Dear Dr. Pierluigi Cocco,

We’re pleased to inform you that your manuscript has been judged scientifically suitable for publication and will be formally accepted for publication once it meets all outstanding technical requirements.

Kind regards,

M. Kariuki Njenga

Academic Editor

PLOS ONE

Additional Editor Comments (optional): The authors have adequately addressed most of the substantive comments from the reviewers.  
---

## [Editor Report · Acceptance letter]

6 Mar 2023

PONE-D-22-29855R1 

COVID-19 and non-Hodgkin’s lymphoma: a common susceptibility pattern? 

Dear Dr. Cocco:

I'm pleased to inform you that your manuscript has been deemed suitable for publication in PLOS ONE. Congratulations! Your manuscript is now with our production department. 

Kind regards, 

on behalf of

Dr. M. Kariuki Njenga 

Academic Editor

PLOS ONE